# IgG3 and IgM Identified as Key to SARS-CoV-2 Neutralization in Convalescent Plasma Pools

**Christina Kober[1]☯, Sandro Manni[2]☯, Svenja Wolff[1], Thomas Barnes[2], Shatanik Mukherjee[1], Thomas Vogel[1], Lea Hoenig[2], Peter Vogel[1], Aaron Hahn[1], Michaela Gerlach[1], Martin Vey[1], Eleonora Widmer[2], Björn Keiner[1], Patrick Schuetz[2], Nathan Roth[2], Uwe Kalina[1]***

**1** Research & Development, CSL Behring Innovation GmbH, Marburg, Germany, **2** Research & Development, CSL Behring, Bern, Switzerland

☯ These authors contributed equally to this work.
* Uwe.Kalina@cslbehring.com

## Abstract

Analysis of convalescent plasma derived from individuals has shown that IgG3 has the most important role in binding to SARS-CoV-2 antigens; however, this has not yet been confirmed in large studies, and the link between binding and neutralization has not been confirmed. By analyzing plasma pools consisting of 247–567 individual convalescent donors, we demonstrated the binding of IgG3 and IgM to Spike-1 protein and the receptor-binding domain correlates strongly with viral neutralization *in vitro*. Furthermore, despite accounting for only approximately 12% of total immunoglobulin mass, collectively IgG3 and IgM account for approximately 80% of the total neutralization. This may have important implications for the development of potent therapies for COVID-19, as it indicates that hyperimmune globulins or convalescent plasma donations with high IgG3 concentrations may be a highly efficacious therapy.

**Data Availability Statement:** All relevant data are within the manuscript and its Supporting Information files.

## Introduction

Coronavirus Disease 2019 (COVID-19), triggered by the severe acute respiratory syndrome coronavirus 2 (SARS-CoV-2), has caused a pandemic with enormous consequences for patients, public health systems and global economics [1]. The humoral response to SARS-CoV-2 has not yet been fully elucidated; understanding the antibody response could pave the way for effective plasma-derived treatment such as hyperimmune globulin and convalescent plasma therapies [2].

Neutralizing antibodies have been shown to disrupt the binding of the SARS-CoV-2 surface spike protein to the angiotensin-converting enzyme 2 (ACE2) [3]. However, data relating the isotypes and subclasses of the antibodies generated in response to SARS-CoV-2 antigens, and their neutralization capability have not yet been investigated in detail.

Amanat *et al.* reported a method to analyze antibodies that can specifically bind the spike protein of the SARS-CoV-2 virus, including immunoglobulin (Ig) sub- and isotype distribution [4]. The development of this assay made serological investigations and large-scale studies

**Funding:** The study was funded by CSL Behring GmbH. The funder provided support in the form of salaries for authors [ Christina Kober, Sandro Manni, Svenja Wolff, Thomas Barnes, Shatanik Mukherjee, Thomas Vogel, Lea Hoenig, Peter Vogel, Aaron Hahn, Michaela Gerlach, Martin Vey, Eleonora Widmer, Björn Keiner, Patrick Schuetz, Nathan Roth, Uwe Kalina], but did not have any additional role in the study design, data collection and analysis, decision to publish, or preparation of the manuscript. The specific roles of these authors are articulated in the 'author contributions' section.

**Competing interests:** I have read the journal's policy and the authors of this manuscript have the following competing interests: All authors are employees of CSL Behring. This does not alter our adherence to PLOS ONE policies on sharing data and materials

possible. Complementary to polymerase chain reaction methods, which solely detect active infections, it allows for the identification of individuals that have recovered from COVID-19. However, this assay does not detect the neutralizing capabilities of antibody sub- and isotypes, and the correlation between binding and neutralizing antibody sub- and isotypes are not yet known.

In this study, we collected and pooled large numbers of plasma donations from convalescent donors, allowing us to study the serological antibody response to SARS-CoV-2 infection at a population level, abrogating individual variation and temporal biases. We aimed to characterize the spectrum of SARS-CoV-2-specific Ig classes (IgG1–4, IgM, IgA) in seven such plasma pools, in terms of viral antigen-binding and contribution to neutralization.

# Material and methods

## Patients and specimens

Human plasma donations, which are part of plasma pools and are analyzed as individual donors, were obtained from varied sources (CSL Plasma donation center network US and Italy, Australian Red Cross Lifeblood, Universitätsklinikum Freiburg, Germany).

Our donors were individuals recovered from a recent illness diagnosed as COVID-19 recruited to provide Source Plasma for further manufacturing into an immune globulin product with high titers of SARS-CoV-2 antibody. At each of their visits at which a sample was provided, a plasmapheresis collection of between 198 and 884 mL was performed. Subjects gave informed consent at the outset of the process for samples of their collection to be tested for various parameters. All individual convalescent plasma donations were tested for NC-binding IgG levels using the high-throughput Abbott Architect System according to the manufacturer's instructions. Donations above the positive cut off (1.4) were classified as convalescent donations and were used in respective convalescent plasma pools or analyzed directly in the study.

Six convalescent plasma pools consisting of 247–273 individual donations were collected by the CSL Plasma donation center network throughout the US. All donations that contributed to the six US plasma pools were collected between March 31, 2020 and June 23, 2020. Fifteen donations were collected in March, 121 in April, 466 in May, and 923 in June 2020. The PCR-positive test date was available for 305 donors, and for these donors, the mean period of convalescent plasma collection after a positive PCR test was 57 ± 19 days.

One convalescent plasma pool with 567 donations was collected by the Australian Red Cross Lifeblood between May 2020 and June 2020 at least 28 days after symptom resolution.

Details with respect to the 7 convalescent plasma pools are summarized in S1 Table.

Three pre-pandemic plasma pools (from Italy and US; generated on 19 January 2019, 27 January 2019 and 21 April 2020) consisting of 10,086, 13,341 and 4,503 donations, were used as negative controls termed as pre-pandemic pools.

## Cell cultures

Vero CCL-81 cells (CCL-81, lot 70016956) were obtained from ATCC. The identity of the cell line as well as absence of mycoplasma contamination are warranted by ATCC as stated in the certificate of analysis provided. Additionally, contamination with mycoplasma was excluded by in-house testing using MycoAlert Mycoplasma Detection Assay (Lonza). Cells were cultivated in Dulbecco's Modified Eagle Medium (DMEM; Gibco) supplemented with 5% fetal calf serum (FCS), 2 mM L-glutamine, 50 U/mL penicillin and 50 μg/ml streptomycin at 37°C and 5% $CO_2$.

## Characterization of convalescent plasma pools using pre-coated ELISA plates (EUROIMMUN)

For analysis of the convalescent plasma pools, an additional ELISA was performed using pre-coated ELISA plates with the recombinant domain of the S1 protein provided with anti-SARS-CoV-2 IgG ELISA Kit (EUROIMMUN; EI 2606–9620). For characterization of convalescent pools, the assay was performed according to the manufacturer's instructions. In brief, samples were tested in 1:101 dilution and the ratio of sample OD and calibrator OD was calculated. For the modified protocol, serial dilutions were tested starting with 1:50 or 1:100. The secondary antibodies against IgG1–IgG4 and IgA (all 1:1000) and IgM (1:50) used for the self-coated plates were also applied here. Prior to assay IgM, samples were pre-treated with EURO-SORB IgG/RF-absorbens (EUROIMMUN) according to manufacturer's instructions by incubation of the samples for 15 min.

## ELISA protocol for binding analysis against different viral antigens

The protocol was developed with Siemens Healthineers and contains the following steps: 96-microwell plates (Thermo Fisher, Typ B U8, Nr. 478310, LOT 155680) were coated with nucleocapsid (Icosagen, P-301-100, 1.75 μg/ml), Spike S1 (Icosagen, P-305-100, 2.0 μg/ml), RBD2 (Icosagen, P-305-107, 2.0 μg/ml) respectively. The plates were washed, dried and stored at 2–8˚C until use. Plasma samples were stored at −80˚C and thawed at 37˚C for 10–15 min. Serial dilutions (starting with 1:50 or 1:100 dilution) of individual convalescent plasma samples and plasma pools were prepared with sample buffer (Siemens, Sample Buffer POD, OWBE). Each sample dilution (100 μL) was applied to the ELISA plate and incubated for 30 min at 37˚C. Plates were washed three times with 300 μL wash buffer (phosphate buffered saline [PBS] with 0.05% TWEEN® 20, pH 7.4; Sigma-Aldrich). For the IgM assay, plates were washed four times using washing solution (Siemens, Washing Solution POD, OSEW). Afterwards plates were incubated for 30 min at 37˚C with horse radish peroxidase-conjugated secondary antibodies diluted in a conjugate buffer (Siemens, conjugate buffer microbiol, OUWW). The following secondary antibodies were applied: mouse anti-human IgG1 Fc antibody (Invitrogen, MH1715, 1:5000), mouse anti-human IgG3 (Hinge region) antibody (Invitrogen, MA5-16718; 1:5000), mouse anti-human IgG2 antibody (Invitrogen, MH1722; 1:5000), mouse anti-human IgG4 Fc antibody (Invitrogen, A-10654; 1:5000), anti-human IgA (α-chain specific) antibody (Sigma-Aldrich, A0295; 1:5000) and anti-human IgM/POD antibody (Siemens, Lot 424004; 1:50). After washing the plates three/four times, 3,3',5,5'-tetramethylbenzidine (TMB substrate; Siemens, OUVG/ OUVF) was added and incubated for 30 min at room temperature (RT). Stop solution (Siemens, OSFA) was added and the optical density (OD) measured at 450 nm. Prior to assay of IgM, the samples were pre-treated with RF-Absorbens (Siemens, OUCG15) for 15 min.

## Depletion experiments

Plasma pool samples were selectively depleted for either IgG1 and IgG3 combined, IgG3, IgG4, IgG1–4, IgM or IgA using Capture Select™ and POROS™ affinity matrices purchased from Thermo (#195289010, #194288005, #191303005, #191304005, #290010, #1943712250). The IgG1 affinity Capture Select™ resin (#191303005) showed affinity for both IgG1 and IgG3 and was used to deplete simultaneously IgG1 and IgG3 from plasma samples. Affinity resin (350 μL) was washed twice with 800 μL PBS. The washed resin was then incubated with 500 μL of convalescent plasma pool sample for 3 h at 4˚C on an overhead shaker. Ig depleted plasma supernatant was recovered and stored at −70˚C for further analysis or assessment of

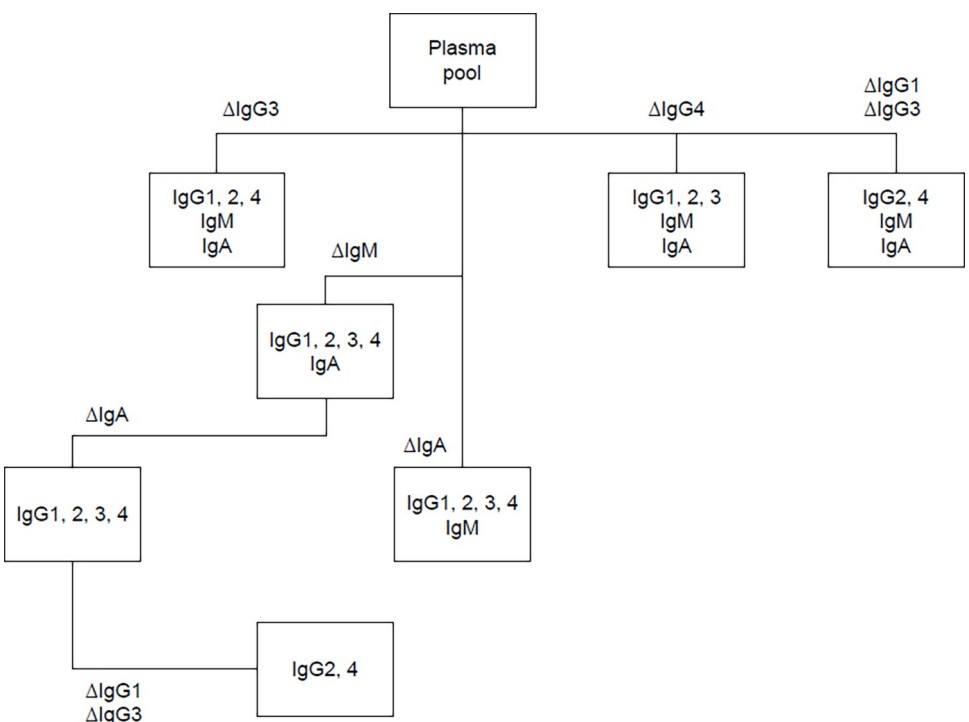

**Fig 1. Experimental depletion workflow of convalescent plasma pool.** Δ indicates removal of specific Ig class.

SARS-CoV-2 neutralization capacity. Multiple Ig class depletions were performed in a sequential way by repeating the above procedure with appropriate resins (Fig 1). Regeneration of the resin was executed according to manufacturer's instructions. Depletion experiments were performed in triplicates. Neutralization capacity of the individually depleted convalescent plasma pools with remaining specific active Ig classes were analyzed in a neutralization assay with live SARS-CoV-2 virus (described below in the neutralization assay section) in triplicates. Neutralization titers per mL (AU [arbitrary unit]/mL ± SD) (Table 1) were determined and used to calculate individual percentage contributions to neutralization. Neutralization titer values for depleted plasma pool samples were normalized against the total Ig content (sum of all Ig classes) and adjusted for an individual Ig mass loss caused by the use of individual resins in the depletion process. The Ig mass loss was assessed using quantitative Ig class ELISAs (described below in the quantitative Ig class ELISAs section). The individual absolute and percentage contributions of SARS-CoV2 neutralizations were calculated based on absolute Ig class content in depleted plasma pool samples.

## Western Blot (WB) analysis

Completeness of IgG class depletion was assessed by WB analysis using specific antibodies against IgG1–IgG4 (The Binding Site, #AU006, #AU007, #AU008, #AU009). Primary sheep anti-human IgG class antibodies were diluted 1:5000 in Odyssey intercept (PBS) blocking buffer (LI-COR, #927–70001) and incubated with the WB membrane for 1 h on a shaker at RT. Blots were then washed 3 x 10 min with tris-buffered saline-Tween and incubated with a goat anti-human infrared-labelled secondary antibody (LI-COR IRDye® 800 CW, #926–32232). WB data was acquired using a LI-COR CLx-Odyssey infrared imaging system (S1 Fig).

**Table 1. Summary of individually depleted plasma pool samples and results of neutralization of SARS-CoV-2.**

| Plasma pool | Depleted Ig-fraction | Active Ig-fraction | SARS-CoV-2 neutralization (AU/mL ± SD) | SARS-CoV-2 neutralization Ig-depleted fraction (%-contribution and range in []) |
|---|---|---|---|---|
| Pre-pandemic | n/a | IgG1, IgG2, IgG3, IgG4, IgM, IgA | <LoQ | n/a |
| Convalescent | n/a | IgG1, IgG2, IgG3, IgG4, IgM, IgA | 119 | n/a |
| Convalescent | IgG3 | IgG1, IgG2, IgG4, IgM, IgA | 52.8 ± 1.6 | 42.2 [40.4–44.0] |
| Convalescent | IgG1, IgG3 | IgG2, IgG4, IgM, IgA | 19.1 ± 2.9 | 57.7 [51.9–63.8] |
| Convalescent | IgG1* | IgG2, IgG3, IgG4, IgM, IgA | n/a | 15.5 [11.5–19.5] |
| Convalescent | IgG4 | IgG1, IgG2, IgG3, IgM, IgA | 103.4 ± 3.0 | <LoQ |
| Convalescent | IgM | IgG1, IgG2, IgG3, IgG4, IgA | 53.0 ± 5.6 | 37.5 [30.9–44.1] |
| Convalescent | IgA | IgG1, IgG2, IgG3, IgG4, IgM | 69.4 ± 1.2 | 7.8 [6.2–9.4] |
| Convalescent | IgG1, IgG3, IgM, IgA | IgG2, IgG4 | <LoQ | <LoQ |

*Theoretically calculated from individual experimental values.

AU, arbitrary units; Ig, immunoglobulin; LoQ, limit of quantitation; n/a, not applicable; NT, neutralization titer; SARS-CoV-2, severe acute respiratory syndrome coronavirus 2; SD, standard deviation (n = 3–4).

## Quantitative Ig class ELISAs

Completeness of IgG1–4, IgM and IgA depletion was further assessed quantitatively using an IgG subclass ELISA (Thermo, #991000) according to the manufacturer's instructions and our in-house IgM and IgA ELISA according to CSL Behring's proprietary standard operating procedure (SOP; S2A and S2B Fig). Mass loss of Igs originating from the depletion workflow were accounted for in the final calculation of neutralization titers of individual Ig classes.

## Nephelometric assay

Ig classes in plasma pools used for depletion experiments were quantified using our in-house nephelometric method according to manufacturers' instruction (Siemens Healthineers).

## Neutralization assay

Neutralizing antibody titers in plasma were tested according to the method previously described by Schwaiger et al. [5]. Briefly, plasma samples were pre-diluted in culture medium (Dulbecco's Modified Eagle Medium supplemented with heat-inactivated fetal bovine serum). Two-fold serial dilutions of each sample were then mixed with an equal volume of SARS-CoV-2 virus (strain BetaCoV/Germany/BavPat1/2020) diluted to 3.0 $\log^{10}$ TCID50/mL in cell culture medium (eight replicates per dilution). Following incubation of the sample at RT for 2.5 h, the virus-sample mixtures were applied to Vero cells (CCL.81; ATCC) seeded and incubated for 5–7 days. Cells were then assessed for cytopathic effects and the NT50 calculated according to the Spearman-Kärber formula [6, 7]. One particular convalescent plasma donation was defined as an "arbitrary standard", and assayed together with, and in the same fashion as, plasma (both single donation and pooled plasma) samples. This standard was declared as having a potency of 100 AU/mL and was included in every measurement. The potency in AU/mL

of samples was calculated by dividing the NT50 of the sample by the NT50 of the arbitrary standard and multiplying the result by 100. The conversion of NT50/ml to an arbitrary unit (AU/ml) allows to determine the relative content of neutralizing antibodies in a given analyte. Moreover, the intrinsic variability effect in a cell-based assay is eliminated, thus allowing a comparison of results generated in different experiments.

## Quantification and statistical analysis

ELISA data was analyzed using GraphPad Prism (8.1.1) and JMP software (15.1.0). An automated JMP script was created to analyze the antibody binding data by a linear AUC model (Fig 2D). The following criteria were used to calculate the AUC using the linear model: a) between two to six values of the inverse of serial dilutions in the linear range; b) either a 1:50 or 1:100 dilution depending on the analyte of interest. Blank values or control values were subtracted from individual measurements for normalization. Optical density values above 2 were not considered for calculation and the warning limit for R square was set to 0.7. For multivariate analysis, correlations were estimated by restricted maximum likelihood (REML) method. We performed a linear correlation and calculated Pearson's r as an indicator of correlation between NT and ELISA titer [8, 9]. In addition, we performed a non-parametric correlation with a monotonic function and calculated Spearman's Rho [10] to confirm the level of correlation. Both methods showed a similar trend.

Student's t-tests were used to determine statistically significant differences between the pre-pandemic and convalescent plasma pools.

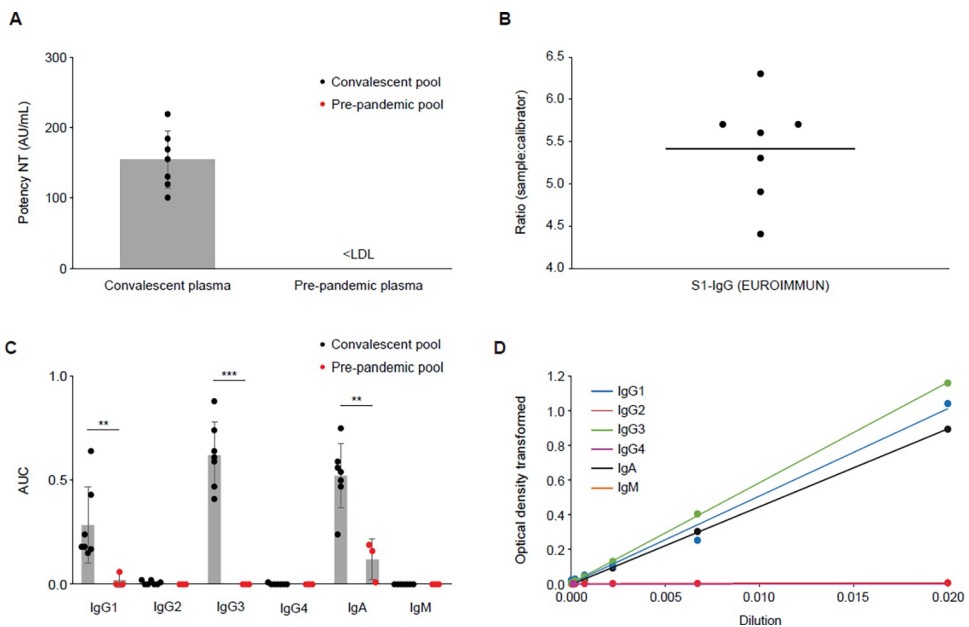

**Fig 2. Serological characterization of investigated plasma pools.** (A) Neutralization titer of convalescent (n = 7) and pre-pandemic (n = 3) plasma pools. Data are shown as mean ± SD. (B) EUROIMMUN testing results (S1-IgG ELISA) for convalescent plasma pools (n = 7). (C) Ig classes of convalescent and pre-pandemic plasma pools against S1 ELISA (modified EUROIMMUN ELISA). Data are shown as mean ± SD. Asterisks indicate the level of statistical significance calculated using the Student's t-test: *p ≤ 0.05, **p ≤ 0.01, and ***p ≤ 0.001. (D) Representative plot for AUC calculation in a single pool sample showing transformed optical density values for different dilutions. AUC, area under the curve; Ig, immunoglobulin; LDL, lower detection limit; NT, neutralization titer; SD, standard deviation.

## Results

### Modified Ig subtype ELISA

Each convalescent pool had a significantly higher SARS-CoV-2-specific neutralization titer (NT) than plasma pools collected prior to the pandemic (Fig 2A). Additionally, each convalescent pool returned a positive result for a diagnostic enzyme-linked immunosorbent assay (ELISA) specific for Spike-1 (S1)-IgG (Fig 2B). The total IgG class distribution was consistent with previous studies [11].

As part of our detailed viral antigen-binding analysis, the S1-binding of Ig classes were investigated using an approach first applied by Amanat *et al.* to individual plasma donors [4]. Calculated area under the curve (AUC) values reflect relative amounts of SARS-CoV-2-specific Ig classes present in the sample (Fig 2C). In accordance with data reported by Amanat *et al.* [4], we observed a similar distribution of IgG classes in our plasma pools: IgG3 exhibited the highest relative abundance (mean AUC 0.62), followed by IgG1 (mean AUC 0.28). Furthermore, IgA showed a high binding signal (mean AUC 0.52), whereas IgM was at baseline level (mean AUC 0.00). In addition, also smaller plasma pools consisting of 12–51 donors were tested showing comparable results with the large pools (S2 Table).

A multi-faceted ELISA method was developed utilizing recombinantly expressed viral antigens with a defined and optimized concentration of (nucleocapsid [NC], S1 and the receptor-binding domain of S1 [S1-RBD]) as capture reagents, and antibodies specific for each Ig class as detection reagents. As this assay is optimized for all Ig classes, simultaneous detection of low levels of IgM in the pools and high levels in some donors which were included in correlation analysis was possible. Applying our in-house data processing method (Fig 2D) revealed that, independent of viral antigen, IgG3 was again the most prominent IgG class (Fig 3A–3C), demonstrating its relevance in SARS-CoV-2 binding.

### Virus neutralization and correlation analysis

The multi-faceted ELISA was applied to a spectrum of individual convalescent plasma donations and to the analyzed convalescent pools, to explore the relationship between specific Ig class binding to SARS-CoV-2 viral antigens and neutralization.

Each binding value was plotted against the NT result for the corresponding sample (Fig 3D–3G and 3H). Interestingly, the strongest NT-binding correlations were found for S1-RBD-IgG3, S1-IgG3, S1-RBD-IgM and S1-IgM (Pearson r = 0.74, 0.71, 0.83, and 0.83, respectively), implicating these classes as important drivers of neutralization (Fig 3H). By contrast, IgA showed a weak correlation to NT for all antigens (Fig 3H).

### Depletion experiment

Over a series of runs, Ig classes were selectively depleted from a convalescent plasma pool comprising 567 plasma donations (Fig 1). Each sample generated was then analyzed for NT (Table 1), as well as the concentration of each Ig class (S1 and S2 Figs).

We determined the contribution of each Ig class using a subtractive approach, correcting for its mass and the mass lost during sample generation, to the total NT of the plasma pool. This revealed that 42% and 38% of the total NT were attributable to IgG3 and IgM (Fig 4), despite collectively only representing 3% and 8% of the total Ig mass, respectively (Fig 4).

By contrast, IgG1 accounted for only 16% of the total NT even though it represented almost half of the total Ig mass (Fig 4). The contribution of IgA to total NT was about half of its contribution to total Ig mass, and IgG2 and IgG4 accounted for a negligible percentage of the total NT (Fig 4).

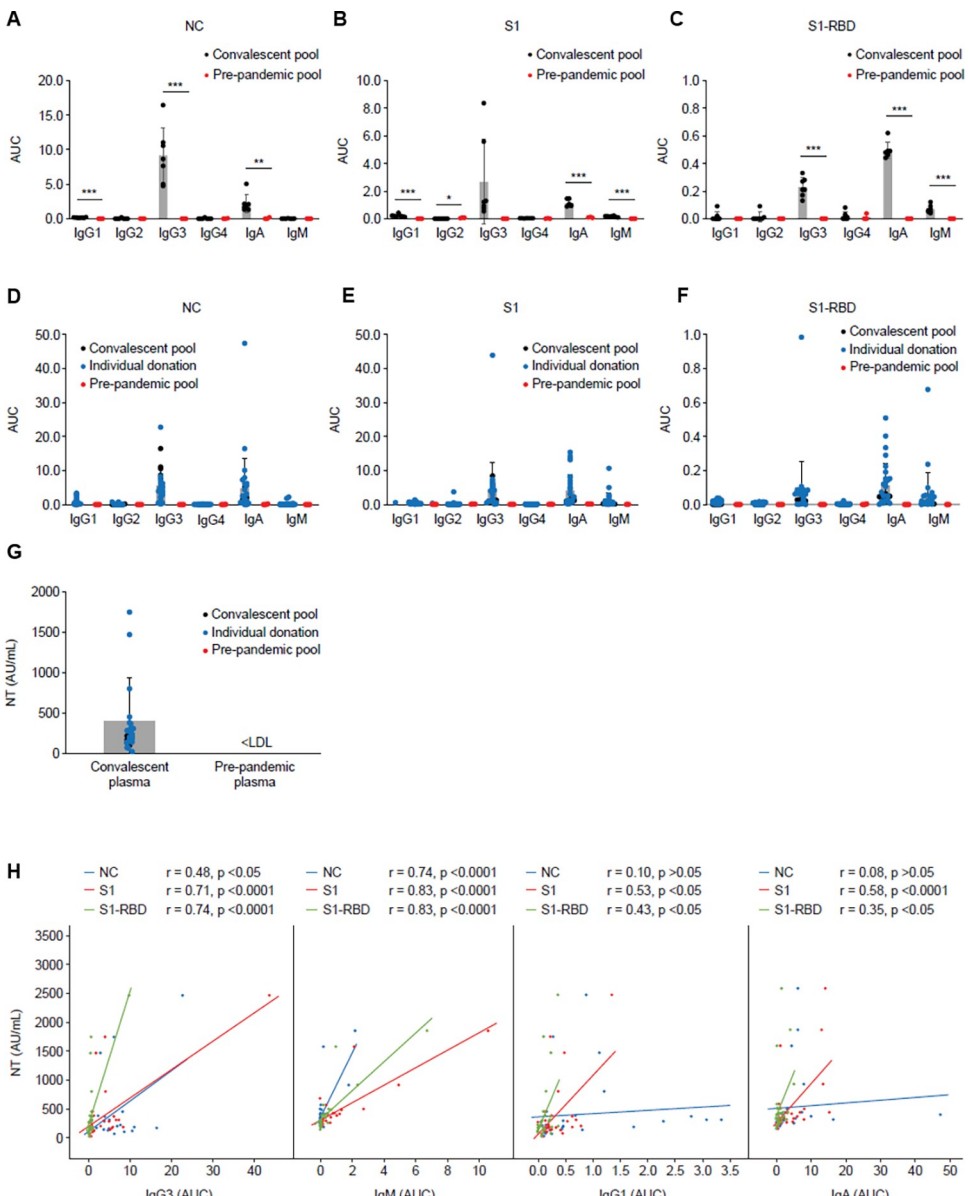

**Fig 3. Binding capability of SARS-CoV-2 specific antibody Ig classes utilizing ELISAs to different viral antigens in selected individual convalescent donors and pools to SARS-CoV-2 viral antigens and correlation analysis.** (A–C) Ig classes of seven convalescent and three pre-pandemic plasma pools against: A) NC; B) S1; C) S1-RBD. Data are shown as mean ± SD. Asterisks indicate the level of statistical significance calculated using the Student's t-test: *p ≤ 0.05, **p ≤ 0.01, and ***p ≤ 0.001. AUC, area under the curve. (D-F) Overlay of specific binding capacity of individual donors, plasma pools, and pre-pandemic plasma pools against D) NC, E) S1, and F) S1-RBD respectively. (G) Neutralizing potency (AU/ml) of pre-pandemic pools, convalescent pools, and individual donors. AU, arbitrary unit; <LDL, below lower detection limit. (H) Pearson's r values are shown as an indicator of correlation between the neutralization potency and IgG3, IgM, IgG1, and IgA with specificity for the NC (blue), S1 (red), or RBD (green) viral antigens. Seven convalescent plasma pools as well as selected individual donors with high, medium, or low NT titers were used for calculation. NC, nucleocapsid; S1, viral antigen S1; S1-RBD, receptor-binding domain of S1; SD, standard deviation.

## Discussion

In this study, we analyzed large convalescent plasma pools to identify which antibody isotypes/ subclasses bind to three different viral antigens, and most effectively neutralize SARS-CoV-2.

**A**                                                    **B**

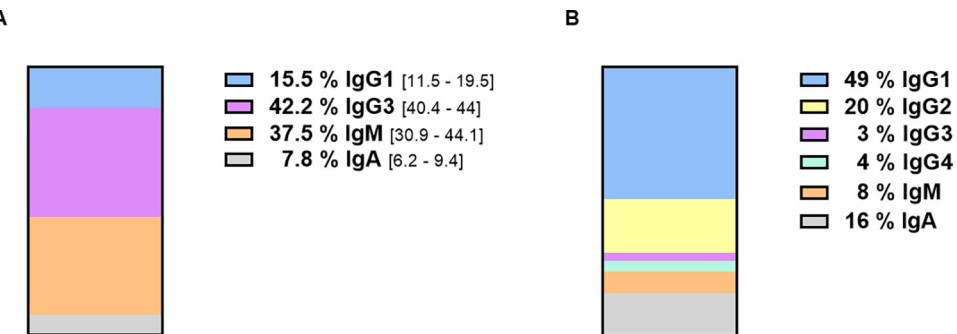

- 15.5 % IgG1 [11.5 - 19.5]
- 42.2 % IgG3 [40.4 - 44]
- 37.5 % IgM [30.9 - 44.1]
- 7.8 % IgA [6.2 - 9.4]

- 49 % IgG1
- 20 % IgG2
- 3 % IgG3
- 4 % IgG4
- 8 % IgM
- 16 % IgA

**Fig 4. Comparison of neutralization capacity against total Ig mass.** (A) Contribution of Ig classes to the neutralization of SARS-CoV-2 (percentage contribution) based on experimental data depicted in Table 1. (B) The abundance of Ig classes as percentage of total Ig in convalescent plasma pool are based on quantitative nephelometric results.

We found that IgG3 and IgM are the most important isotypes/subclasses for virus neutralization and that the strongest correlations are observed against spike S1 and RBD proteins.

This study provides direct attribution of neutralization to Ig classes, namely IgG3 and IgM, for COVID-19. For other enveloped viruses including human immunodeficiency virus type 1, IgG3 has been isolated and shown to be the major contributor to neutralization [12]. It has been postulated that this is due to the relatively long hinge region of IgG3, which allows for greater rotational flexibility and improved capacity to bind multivalent antigens as described by Damelang *et al.* [13]. Our data, generated from two orthogonal methods, implicates S1-RBD-specific IgG3 as critical in SARS-CoV-2 neutralization. This is consistent with the finding by Ju *et al.* that disruption of S1-RBD binding to the ACE2 receptor by IgG prevents viral entry [14].

Collectively, our data indicate that IgM plays a pivotal role in viral neutralization, despite its relatively low abundance. This may be due to the polyvalent nature of IgM and the higher avidity towards antigens compared with divalent IgG and IgA. It has been shown that high avidity of neutralizing antibodies play an important role in humoral response against viral infections. Binding between the receptor-binding domain (RBD) of SARS-CoV-2 and the angiotensin-converting enzyme 2 (ACE2) is of high affinity. In a study from Khatri *et al*, it was concluded that an efficient neutralization of SARS-CoV-2 would require antibodies of high avidity [15]. High avidity IgM immunoglobulins may thus be capable to exert an inhibitory effect already at low concentration. However, additional studies are required to determine if this is the case. This study builds on the findings of Gasser et al., who selectively depleted individual plasma samples of IgA, IgM, or IgG, finding that IgM and IgG play key roles in the neutralization of SARS-CoV-2 [16]. However, while Gasser et al. tested the plasma of 25 individual donors, this study examined large plasma pools, collected from 247–567 donors, which allows for a more comprehensive, population level analysis. A further difference between this study and Gasser et al. is the role of the different antibody isotypes; both studies found IgM and IgG to be important for neutralizing SARS-CoV-2, but Gasser et al. reported that IgM had a greater role in neutralizing SARS-CoV-2, whereas this study found IgG3 to be more important. Studies such as Prévost et al. and Lui et al. have reported that IgM neutralizing antibodies are the first to be detected, but decrease over time, whereas IgG neutralizing antibodies appear later and remain relatively stable over time [17, 18]. This is consistent with data on antibody maturation and seroconversion over the course of infection as reported for other viral infections such as HCoV-229E, MERS-CoV, and SARS-CoV-1 [19].

In contrast to our study, in which IgG3 was found to be the most important IgG subclass for neutralizing SARS-CoV-2, a number of smaller studies have found that IgG1 plays a more important role [20–22]. Klingler et al. analyzed samples from 29 convalescent donors and found that IgM and IgG1 had significant neutralizing activity against the spike and RBD of SARS-CoV-2; whereas IgG3 had significant neutralizing activity against the spike protein, but not the RBD [23]. Luo et al. analyzed data from 63 convalescent donors who experienced severe, moderate, mild and asymptomatic COVID-19. Luo et al. found that both IgG1 and IgG3 were key to the humoral immune response [24]. In addition, Mazzini *et al*. analyzed 181 human serum samples using commercial and in-house ELISA assays, and found that IgG1 and IgG3 had the strongest reactivity to the SARSCoV-2 antigens spike S1 and spike-RBD [25].

The reasons for the disparities regarding the role of IgG1 and IgG3 are unclear. They may be related to the temporal model of affinity maturation, with the IgG3 response earlier, with a switch to a higher affinity IgG1 antibody response later [26]. Also, it has been reported that the time-point of donation as well as disease severity are factors influencing the outcome of the antibody profile [27]. Alternatively, the variations could be due to methodological differences in how the neutralization assays were conducted.

IgG2 and IgG4, the other subclasses of IgG did not contribute to either the binding of the tested SARS-CoV-2 antigens or the neutralization of the virus similar to the observation in other studies [17, 28, 29]. A recent study showed that IgG4 in fact is a marker for mortality with patients having higher anti-RBD IgG4 levels dying during 8–14 and 15–21 days [30]. This suggests that the antibody subclass elicitation to SARS-CoV-2 is broad and correlates strongly to the characteristics of the donor population and sample collection time. Additionally, the IgG subclasses also have variable binding and neutralizing capabilities against protein antigens [31]. IgG1 and IgG3 responses are mostly generated against soluble and membrane protein antigens whereas IgG2 and IgG4 arise against bacterial capsular polysaccharide and allergen molecules respectively. Interestingly, it has been shown that expressing the same spike protein-binding monoclonal antibody in human-IgG1-4 background resulted in 5-fold superior binding affinity and 50-fold superior neutralization capacity in case of IgG3 over other subclasses. The authors postulate that IgG3 elicits a superior binding and neutralization effect against SARS-CoV-2 in an avidity-dependent manner via cross-linking the spike protein on the viral surface [32].

Interestingly, the high binding levels of IgA do not translate to a strong neutralizing activity. We rather see a moderate level of correlation between antigen-binding and neutralization activity of IgA compared to IgG3 and IgM in the tested samples. In fact, similar correlation coefficients were observed for IgA and IgG1. The Ig-depletion study also indicated presence of low to moderate levels of neutralization contribution by IgA. This contradicts findings from Chen *et al*. showing strong correlation between S1-specific and ECD-specific IgA responses and neutralization activity in non-severe patients [33]. However, the neutralization activities of the convalescent sera were shown to significantly decline during the period between 21 days to 28 days after hospital discharge along with a substantial drop in RBD-specific IgA response. These findings are in agreement with Sterlin *et al* reporting that the early SARS-CoV-2 specific humoral responses were strongly driven by IgA antibodies. In addition, peripheral expansion of IgA plasmablasts was detected shortly after symptom onset and peaked during the third week of the diseases at around day 22 [34]. Serum IgA concentrations decreased notably 1 month after onset of symptoms. These findings demonstrate that IgA may play a more important role in the early phases of infection. The difference with our findings may be due to time points of plasma collection. Our plasma donations were collected from convalescent donors who were symptom free for at least 28 days. It was shown that at this stage (at around day 15–28) IgA and IgG peaked. However, IgA levels start to wane whereas IgG remained stable [35].

While we have so far investigated only viral neutralization by immunoglobulins, it will be critical to understand the impact of Ig classes in mediating Fc effector functions such as complement activation and Fc gamma receptor binding in the context of SARS-CoV-2 *in vivo*. Studies that have profiled the Fc receptor response in SARS-CoV-2 infection have reported that immunoglobulins have an important role in mediating inflammatory effects [36, 37].

A limitation of this study is that the convalescent plasma pools used for this analysis were sourced from the USA and Australia, and so may not be representative of all patient populations. Additionally, this data does not characterize the antibody response over time, which would require long-term studies. In this study, the plasma pools collected were not stratified by time-point after symptom remission, which is a further limitation. However, donations were collected in a similar timeframe (at least 30 days post-positive PCR test). Another limitation of this study is that efficacy has only been shown *in vitro*.

This work may have important implications for the development of potent therapies for COVID-19. Efficacy trials investigating convalescent plasma have delivered ambiguous results [38, 39]; however, therapies providing high IgG concentrations [40, 41], particularly those high in IgG3, may prove to be efficacious. In contrast to convalescent plasma treatment, hyperimmune globulins, as manufactured products, contain consistent levels of concentrated SARS-CoV-2 specific antibodies, and offer several benefits over convalescent plasma transfusion therapy including improved safety profile, longer shelf-life stability, and manufacturing scalability.

## Supporting information

**S1 Fig. Western blot analysis showing the quantity of Ig classes (IgG1, IgG2, IgG3 and IgG4) retained in a representative set of depleted convalescent plasma pool used for SARS-CoV-2 neutralization.** Δ indicates removal of specific Ig classes. IB, immunoblot; M, marker.
(TIF)

**S2 Fig.** ELISA of Ig classes in depleted plasma pool samples used for SARS-CoV-2 neutralization (A) IgA; (B) IgM; (C) IgG1–4. Bars are mean ± SD. Δ indicates removal of specific Ig classes.
(TIF)

**S1 Table. Overview on analyzed large convalescent plasma pools including pooling time-point, volume, and country of collection.**
(TIF)

**S2 Table. Anti-SARS-CoV-2 S1 antibody levels in smaller sized convalescent plasma pools consisting of 12–51 donors.**
(TIF)

## Acknowledgments

The authors would like to thank their colleagues Sara Stinca, Michèle Werren, Claudia Hadorn, Roland Weber, Harald Steller, Michelle Williams, Robin Jenness, Catherine Wright, Jessica Shan, Supavadee Amatayakul-Chantler for their contributions to this study. The contribution of Siemens Healthineers, most notably Herbert Schwarz, is gratefully acknowledged. His team supported the development (e.g. coating of the ELISA plates) and measurement of the in-house ELISA using Icosagen antigens. We would like to thank Meridian HealthComms Ltd. for editorial assistance.

## Author Contributions

**Conceptualization:** Christina Kober, Sandro Manni, Svenja Wolff.

**Formal analysis:** Christina Kober, Sandro Manni, Svenja Wolff, Shatanik Mukherjee, Peter Vogel.

**Investigation:** Christina Kober, Sandro Manni.

**Methodology:** Christina Kober, Sandro Manni, Shatanik Mukherjee, Thomas Vogel, Lea Hoenig.

**Supervision:** Martin Vey, Eleonora Widmer, Björn Keiner, Patrick Schuetz, Nathan Roth, Uwe Kalina.

**Writing – original draft:** Christina Kober, Sandro Manni, Thomas Barnes, Shatanik Mukherjee, Aaron Hahn.

**Writing – review & editing:** Christina Kober, Sandro Manni, Svenja Wolff, Thomas Barnes, Shatanik Mukherjee, Thomas Vogel, Lea Hoenig, Peter Vogel, Aaron Hahn, Michaela Gerlach, Martin Vey, Eleonora Widmer, Björn Keiner, Patrick Schuetz, Nathan Roth, Uwe Kalina.

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
