## [Decision Letter · Decision Letter 0]

11 Oct 2021

PONE-D-21-26634IgG3 and IgM Identified as Key to SARS-CoV-2 Neutralization in Convalescent Plasma PoolsPLOS ONE

Dear Dr. Kalina

Thank you for submitting your manuscript to PLOS ONE. After careful consideration, we feel that it has merit but does not fully meet PLOS ONE’s publication criteria as it currently stands. Therefore, we invite you to submit a revised version of the manuscript that addresses the points raised during the review process. 

Two reviewers have assessed your manuscript. The reviewers consider your manuscript of interest, as addressing a problem of clinical importance. However, they expressed a series of concerns that need to be carefully addressed before we can consider your manuscript for publication. Therefore, we would like to invite you to REVISE your paper. In addition to the reviewer comments, I have also one concern. Sterlin et al [(Science translational medicine. 2021;13(577)] demonstrate that IgA dominates the early neutralizing antibody response to SARS-CoV-2. In your discussion, could you provide explanation to the differences between your and their findings.  

Please submit your revised manuscript by  the Nov 25 2021 11:59PM. If you will need more time than this to complete your revisions, please reply to this message or contact the journal office at plosone@plos.org. Please include the following items when submitting your revised manuscript:A rebuttal letter that responds to each point raised by the academic editor and reviewer(s). You should upload this letter as a separate file labeled 'Response to Reviewers'.A marked-up copy of your manuscript that highlights changes made to the original version. You should upload this as a separate file labeled 'Revised Manuscript with Track Changes'.An unmarked version of your revised paper without tracked changes. You should upload this as a separate file labeled 'Manuscript'.

We look forward to receiving your revised manuscript.

Kind regards,

Gheyath K. Nasrallah

Academic Editor

PLOS ONE

Additional Editor Comments (if provided):

Two reviewers have assessed your manuscript. The reviewers consider your manuscript of interest, as addressing a problem of clinical importance. However, they expressed a series of concerns that need to be carefully addressed before we can consider your manuscript for publication. Therefore, we would like to invite you to REVISE your paper. In addition to the reviewer comments, I have also one concern. Sterlin et al [(Science translational medicine. 2021;13(577)] demonstrate that IgA dominates the early neutralizing antibody response to SARS-CoV-2. In your discussion, could you provide explanation to the differences between your and their findings.

Journal Requirements:

2. Thank you for stating the following in the Competing Interests/Financial Disclosure* (delete as necessary) section:

“I have read the journal's policy and the authors of this manuscript have the following competing interests: All authors are employees of CSL Behring.”

We note that one or more of the authors are employed by a commercial company CSL Behring    1.    Please provide an amended Funding Statement declaring this commercial affiliation, as well as a statement regarding the Role of Funders in your study. If the funding organization did not play a role in the study design, data collection and analysis, decision to publish, or preparation of the manuscript and only provided financial support in the form of authors' salaries and/or research materials, please review your statements relating to the author contributions, and ensure you have specifically and accurately indicated the role(s) that these authors had in your study. You can update author roles in the Author Contributions section of the online submission form.

“The study was funded by CSL Behring GmbH.”

“The authors would like to acknowledge the following CSL employees for logistical support and laboratory analysis: Sara Stinca, Michèle Werren, Claudia Hadorn, Michaela Gerlach, Roland Weber, Harald Steller, Michelle Williams, Robin Jenness, Catherine Wright, Jessica Shan, Supavadee Amatayakul-Chantler.”

“The study was funded by CSL Behring GmbH.”

Reviewers' comments:

Reviewer's Responses to Questions

**Comments to the Author**

1. Is the manuscript technically sound, and do the data support the conclusions?

Reviewer #1: Yes

Reviewer #2: Yes

2. Has the statistical analysis been performed appropriately and rigorously? 

Reviewer #1: Yes

Reviewer #2: Yes

3. Have the authors made all data underlying the findings in their manuscript fully available?

Reviewer #1: Yes

Reviewer #2: Yes

4. Is the manuscript presented in an intelligible fashion and written in standard English?

Reviewer #1: Yes

Reviewer #2: Yes

5. Review Comments to the Author

Reviewer #1: The manuscript in hand presents results on the potential role of IgM and IgG3, taken from convalescent plasma pools derived from individuals who recovered from the disease, to neutralize the virus in vitro.

The following remarks need your attention:

- What is the red line in Fig 1D? couldn’t delineate/differentiate colors in Fig. 1D!

- In page 4, the 1st paragraph speaks about ALL convalescent pools and how candidates were selected, and their plasma tested. In paragraph 2 of same page the authors discuss six pools in addition to the Australian pool. My assumption was that ALL pools were treated the same way in terms of having to wait for at least 28 days and not only the Australian pool, is this correct? - -

- The authors used pooled “donations”, were these donations similar to blood bank donations? If so, don’t you think that pooling 200-300 donations into one pool was too much? Such large pooling “system” has the potential to hide any Ig’s variation, why not use smaller number of donations (e.g. 20-50 donations) in each pool for more “realistic” approach? Also, What is the volume of each pool (i.e. volume range)? and how pooling was done?

- 1st and 2nd paragraphs of page 4 should be re-written for uniformity.

- Isn’t there a contradiction between Fig 1C, Fig 1D and Table 1 regarding IgM result outcomes? figure 1C shows background amounts of anti-S1 IgM antibodies which will not make a difference if depleted, whereas Table 1 speaks about significant contribution of IgM to the neutralization process (also line 200)!

- Lines 235-239: results of Fig 2 (A-C) show minimal or baseline amounts of IgM which contradicts what was exhibited in Fig 2D (specifically the IgM graph).

Reviewer #2: The manuscript IgG3 and IgM Identified as Key to SARS-CoV-2 Neutralization in Convalescent Plasma Pools may have important implications for the development of potent therapies for COVID-19. By analyzing plasma pools consisting of 247–567 individual convalescent donors, the authors demonstrated the binding of IgG3 and IgM to Spike-1 protein and the receptor-binding domain correlates strongly with viral neutralization in vitro. Furthermore, despite accounting for only approximately 12% of total immunoglobulin mass, collectively IgG3 and IgM account for approximately 80% of the total neutralization. The hyperimmune globulins with high IgG3 and IgM concentrations may be a highly efficacious therapy. My recommendations suggest minor revisions on this manuscript to be accepted in Plos One.

Some point should be addressed before resubmission:

1) Line 38 - Determine if there is an approval in the bioethics committee at this stage of the methodology. This was not commented in the manuscript.

2) Line 42 and others - In my way of understanding all the supplementary figures could be placed as the main ones. These figures contain a lot of relevant information fundamental to the consistency of the article.

3) Line 103 - to be more didactic about the presentation of antibody depletion, I find it interesting to put the scheme together with the methodology.

4) Line 142 – insert the reference of this methodology used by the group.

5) Line 155 - add reference too

6) Line 267 - Conversely, do the authors have any postulation or comment regarding the low efficiency of neutralization in IgG2 and IgG4?

6. PLOS authors have the option to publish the peer review history of their article (what does this mean?). If published, this will include your full peer review and any attached files.

Reviewer #1: No

Reviewer #2: **Yes: **Carlos Prudencio

---

## [Author Response · Author response to Decision Letter 0]

9 Nov 2021

A point-by-point Response and description of the changes that have been made in the revised manuscript is provided in the attached document "Response to Reviewers (Rebuttal)" and the Manuscript including track changes.

---

## [Decision Letter · Decision Letter 1]

9 Dec 2021

PONE-D-21-26634R1IgG3 and IgM Identified as Key to SARS-CoV-2 Neutralization in Convalescent Plasma PoolsPLOS ONE

Dear Dr. Kalina

We are very close to acceptance. However, reviewer 2 has raised very important comment that needs to be address. I agree with his comment. To support your answer please refer to 

D Sterlin, A Mathian, M Miyara, et al. Iga dominates the early neutralizing antibody response to sars-cov-2. Science translational medicine 2021; 13(577).<o:p></o:p>

We look forward to receiving your revised manuscript.

Kind regards,

Gheyath K. Nasrallah

Academic Editor

PLOS ONE

Journal Requirements:

Reviewers' comments:

Reviewer's Responses to Questions

**Comments to the Author**

1. If the authors have adequately addressed your comments raised in a previous round of review and you feel that this manuscript is now acceptable for publication, you may indicate that here to bypass the “Comments to the Author” section, enter your conflict of interest statement in the “Confidential to Editor” section, and submit your "Accept" recommendation.

Reviewer #1: (No Response)

Reviewer #2: All comments have been addressed

2. Is the manuscript technically sound, and do the data support the conclusions?

Reviewer #1: Yes

Reviewer #2: Yes

3. Has the statistical analysis been performed appropriately and rigorously? 

Reviewer #1: Yes

Reviewer #2: Yes

4. Have the authors made all data underlying the findings in their manuscript fully available?

Reviewer #1: Yes

Reviewer #2: Yes

5. Is the manuscript presented in an intelligible fashion and written in standard English?

Reviewer #1: Yes

Reviewer #2: Yes

6. Review Comments to the Author

Reviewer #1: Since the last review, the authors have adequately addressed issues raised by the reviewer. My remaining issue is the IgA! In their rebuttal, the authors attached a table (Supplemental S2 Table: Anti-SARS-CoV-2 S1 antibody levels in smaller sized convalescent plasma pools consisting of 12-51 donors); the table clearly shows the dominance of IgA, as measured by AUC, as neutralizing Ab, although the authors still maintain that IgA is of less importance in this regard after 28 days! The pooled plasma used in the study is about 28 days post infection, and I understand that IgA was shown, in previous studies, of less importance and declines around a month post infection, your data says, in several places a different outcome! The question remains: Don't you think that IgA should still be considered as a valid option? The rationale for conducting the study is to justify, in the future, using IgG3 and IgM as passive therapeutic Abs. Don't you think that IgA would be a more valid candidate over IgG's and IgM especially in severe lung disease since it is a secretory Ab?

Reviewer #2: In this manuscript, the authors presented important informations for the development of potent therapies for COVID-19, as it indicates that hyperimmune globulins or convalescent plasma donations with high IgG3 concentrations may be a highly efficacious therapy. The authors have adequately addressed my comments and concerns raised.

7. PLOS authors have the option to publish the peer review history of their article (what does this mean?). If published, this will include your full peer review and any attached files.

Reviewer #1: No

Reviewer #2: **Yes: **Carlos Roberto Prudencio

---

## [Author Response · Author response to Decision Letter 1]

15 Dec 2021

Reviewer #1: Since the last review, the authors have adequately addressed issues raised by the reviewer. My remaining issue is the IgA! In their rebuttal, the authors attached a table (Supplemental S2 Table: Anti-SARS-CoV-2 S1 antibody levels in smaller sized convalescent plasma pools consisting of 12-51 donors); the table clearly shows the dominance of IgA, as measured by AUC, as neutralizing Ab, although the authors still maintain that IgA is of less importance in this regard after 28 days! The pooled plasma used in the study is about 28 days post infection, and I understand that IgA was shown, in previous studies, of less importance and declines around a month post infection, your data says, in several places a different outcome! The question remains: Don't you think that IgA should still be considered as a valid option? The rationale for conducting the study is to justify, in the future, using IgG3 and IgM as passive therapeutic Abs. Don't you think that IgA would be a more valid candidate over IgG's and IgM especially in severe lung disease since it is a secretory Ab?

We thank the reviewer for the critical observation. Indeed, we were also surprised by the high binding activity of IgA. However, we would like to point out that although the Table S2 and Fig. 2C, D show a high AUC value for IgA (both in large and small plasma pools), it reflects only Anti-S1-binding activity and not neutralization of the SARS-CoV-2 virus. In our study, we have linked antigen-binding and ascertained relative contribution of each Ig classes towards neutralizing activity by two methods: 

1. correlation analysis between neutralization and antigen binding

2. experimentally via Ig-class depletion

First, we found only a moderate correlation between IgA antigen-binding and neutralization activity with correlation coefficients IgG3>IgM>IgG1>IgA. Second, we found with our depletion study again the same trend of contribution towards neutralization with IgG3>IgM>IgG1>IgA. These data unambiguously show that among the major Ig classes of interest, IgA to be of least neutralizing potency in the tested samples. Additionally, the plasma samples used in the study were from convalescent individuals at >28 days post-recovery phase where IgA levels have been shown to wane (Sterlin et al.). A longitudinal examination of SARS-CoV-2 neutralizing IgA and IgG levels can provide definitive answers but this was not in the purview of the current study.

We however realize that for a respiratory infectious disease, especially the secreted form of IgA could be a means of early defense against pathogens and therefore a potential therapeutic option. But secretory IgA predominant in mucosal tissue is dimeric and was shown to have different neutralizing activity from the monomeric plasma IgA (Sterlin et al.). Since, we have not tested the secretory form of IgA in this study, we cannot comment on that. 

As our aim was to create a passive immunotherapy for IV application, we focused on plasma samples which clearly indicated IgG3 and IgM to be the prime drivers of neutralization in plasma. This of course does not rule out mucosal administration of IgA to prevent infection with SARS-CoV-2 but to draw this conclusion from our data would be speculative at best.

To make this point clearer, we have added the following sentences in the manuscript in sentence 341.

 „ Interestingly, the high binding levels of IgA do not translate to a strong neutralizing activity. We rather see a moderate level of correlation between antigen-binding and neutralization activity of IgA compared to IgG3 and IgM in the tested samples. In fact, similar correlation coefficients were observed for IgA and IgG1. The Ig-depletion study also indicated presence of low to moderate levels of neutralization contribution by IgA. “

---

## [Editor Report · Decision Letter 2]

19 Dec 2021

IgG3 and IgM Identified as Key to SARS-CoV-2 Neutralization in Convalescent Plasma Pools

PONE-D-21-26634R2

Dear Dr. Kalina,

We’re pleased to inform you that your manuscript has been judged scientifically suitable for publication and will be formally accepted for publication once it meets all outstanding technical requirements.

Kind regards,

Gheyath K. Nasrallah

Academic Editor

PLOS ONE
---

## [Editor Report · Acceptance letter]

23 Dec 2021

PONE-D-21-26634R2 

IgG3 and IgM Identified as Key to SARS-CoV-2 Neutralization in Convalescent Plasma Pools 

Dear Dr. Kalina:

I'm pleased to inform you that your manuscript has been deemed suitable for publication in PLOS ONE. Congratulations! Your manuscript is now with our production department. 

Kind regards, 

on behalf of

Dr. Gheyath K. Nasrallah 

Academic Editor

PLOS ONE